# *Solanum lycopersicum* (Tomato)-Derived Nanovesicles Accelerate Wound Healing by Eliciting the Migration of Keratinocytes and Fibroblasts

**DOI:** 10.3390/ijms25052452

**Published:** 2024-02-20

**Authors:** Valeria Daniello, Vincenzo De Leo, Maria Lasalvia, Md Niamat Hossain, Annalucia Carbone, Lucia Catucci, Roberto Zefferino, Chiara Ingrosso, Massimo Conese, Sante Di Gioia

**Affiliations:** 1Department of Clinical and Experimental Medicine, University of Foggia, Via Napoli 121, 71122 Foggia, Italy; valeria.daniello@unifg.it (V.D.); maria.lasalvia@unifg.it (M.L.); mdniamat.hossain@unifg.it (M.N.H.); annalucia.carbone@unifg.it (A.C.); sante.digioia@unifg.it (S.D.G.); 2Department of Chemistry, University of Bari “Aldo Moro”, Via Orabona 4, 70126 Bari, Italy; vincenzo.deleo@uniba.it (V.D.L.); lucia.catucci@uniba.it (L.C.); 3Department of Medical and Surgical Sciences, University of Foggia, Via Napoli 121, 71122 Foggia, Italy; roberto.zefferino@unifg.it; 4Institute for Chemical and Physical Processes of National Research Council (CNR-IPCF), S.S. Bari, c/o Department of Chemistry, University of Bari “Aldo Moro”, Via Orabona 4, 70126 Bari, Italy; c.ingrosso@ba.ipcf.cnr.it

**Keywords:** *Solanum lycopersicum*, nanovesicles, wound healing, proliferation, migration, keratinocytes, fibroblasts

## Abstract

Plant-derived nanovesicles have been considered interesting in medicine for their breakthrough biological effects, including those relevant to wound healing. However, tomato-derived nanovesicles (TDNVs) have not been studied for their effects on wound closure yet. TDNVs were isolated from *Solanum lycopersicum* (var. Piccadilly) ripe tomatoes by ultracentrifugation. Extract (collected during the isolation procedure) and NVs (pellet) were characterized by transmission electron microscopy and laser Doppler electrophoresis. Wound healing in the presence of Extract or NVs was analyzed by a scratch assay with monocultures of human keratinocytes (HUKE) or NIH-3T3 mouse fibroblasts. Cell proliferation and migration were studied by MTT and agarose spot assay, respectively. The vesicles in the Extract and NV samples were nanosized with a similar mean diameter of 115 nm and 130 nm, respectively. Both Extract and NVs had already accelerated wound closure of injured HUKE and NIH-3T3 monocultures by 6 h post-injury. Although neither sample exerted a cytotoxic effect on HUKE and NIH-3T3 fibroblasts, they did not augment cell proliferation. NVs and the Extract increased cell migration of both cell types. NVs from tomatoes may accelerate wound healing by increasing keratinocyte and fibroblast migration. These results indicate the potential therapeutic usefulness of TDNVs in the treatment of chronic or hard-to-heal ulcers.

## 1. Introduction

Wound healing is a complex process involving hemostasis, inflammation, proliferation, and migration of stem cells, keratinocytes, fibroblasts, and endothelial cells, as well as collagen deposition and remodeling [1]. Hemostasis is critical to provide at the beginning of the skin wound healing process as a provisional matrix, enriched with cross-linked fibrin, fibronectin, vitronectin, and thrombospondin, which serve as a substrate for migration of keratinocytes (including epidermal stem cells), blood cells, and endothelial cells. Moreover, platelets secrete a wealth of soluble mediators inciting and enduring the following phase, i.e., inflammation. Thus, platelets and leukocytes secrete growth factors and cytokines, which attract and activate inflammatory cells in the injury site [2,3]. While neutrophils are deployed for the debridement of necrotic tissue and removal of the injury cause, especially microorganisms, macrophages are devoted not only to phagocyte cell debris and pathogens but principally to orchestrate keratinocytes, endothelial cells, and fibroblast migration in order to ensure re-epithelization and neoangiogenesis, supporting the formation of the granulation tissue as well as the synthesis of extracellular matrix components by fibroblasts [4,5]. The last phase is characterized by a change in collagen-type deposition (from collagen III to I) and its remodeling due to the balance between deposition and degradation exerted by metalloproteinases, resulting in the acellular and avascular scar. If these phases are not truly coordinated, excess inflammation, altered neovascularization, slowed down re-epithelization, or impairment in remodeling occurs, resulting in a delayed wound healing process due to local and systemic factors, such as hypoxia, nutrition, infection, stress, aging, medications, and chronic diseases [6]. Underhealing wounds, such as nonhealing chronic ulcers, pose a significant healthcare challenge. Clinical examples are represented by chronic and hard-to-heal ulcers of different etiology, including diabetic ulcers, venous ulcers, radiation ulcers, and pressure ulcers, which represent a serious worldwide medical and social problem [3,7]. Current chronic wound therapies are comprised of surgical debridement, the application of various types of dressing, the use of skin grafts, topical formulations, scaffolds, and skin substitutes; however, their employment is limited, warranting the development of safe, efficient, and low-cost novel wound therapies [6,8].

Green medicine therapy strategies are used as a new therapeutic approach for wound healing not only for their effectiveness, reliability, and safety but also for their low cost [9]. Pharmacological targets of plant- and herbal-derived nanostructures include suppressing the production of inflammatory cytokines and inflammatory transduction cascades, reducing oxidative factors and enhancing antioxidative enzymes, and promoting neovascularization and angiogenic pathways [9]. Both mammalian- and plant-derived nanovesicles (PDNVs) are at the forefront of these advancements in wound healing [9,10]. The relevance of PDNVs in interspecies communication is derived from their content in biomolecules, absence of toxicity, and easy internalization by mammalian cells, as well as from their anti-inflammatory and immuno-modulatory properties [11]. PDNVs isolated from plants consist of a lipid bilayer containing proteins, lipids, and microRNAs (miRNAs). They contain molecules absent in mammalian extracellular vesicles (EVs), and their combination collectively grants them unique natural bioactivity, making these vesicles a promising approach for tissue preservation and regeneration owing to their inherent characteristics. PDNVs naturally encapsulate various natural drugs, creating a “protective shield” effect that improves their bioavailability [12]. Moreover, PDNVs possess a good natural penetrability insofar as PDNVs can cross the cell barrier, blood–brain barrier, and skin barrier to protect or repair the tissues [13,14]. It is interesting to point out that PDNVs originating from natural sources and commonly included in our daily diet demonstrate minimal immunogenicity and display excellent biocompatibility [15].

Interestingly, some PDNVs have emerged as a tool for wound healing. The first studies indicated that broccoli-derived NVs could improve the disease state at the level of the intestine in inflammatory bowel disease mouse models by affecting inflammation and promoting pro-healing effects [16,17]. Others then found that wheat-derived exosomes promoted wounds to close more rapidly by increasing keratinocyte, fibroblast, and endothelial cell proliferation and migration, as well as neoangiogenesis [18]. Moreover, grapefruit-derived NVs stimulated keratinocyte migration in a scratch assay and induced tube formation in the human endothelial cell line HUVEC [19]. Pomegranate-derived EVs promoted significant wound healing effects on the colon cell line Caco-2 [20]. *Aloe saponaria*-derived NVs increased human dermal fibroblast proliferation and migration as well as induced the formation of capillary-like tube formation by HUVECs [21]. More recently, ginger-derived NVs were also shown to induce faster wound healing and migration of keratinocytes and HUVEC tube-like structure formation in this last cell line [22].

Tomatoes, a cultivation spread in the Apulia region of Italy, have well-known antioxidant and anti-inflammatory activities of therapeutic components, such as lycopene, ascorbic acid, and α-tocopherol [23]. Tomato fruits produce a diverse range of phytochemicals known for their potential health benefits in humans, including carotenoid pigments like lycopene and β-carotene, glycoalkaloids such as dehydrotomatine, α-tomatine, and esculeoside A, as well as numerous phenolic compounds [24]. Several of these compounds exhibit strong antioxidant properties, effectively scavenging free radicals that can cause damage to DNA, proteins, and membrane lipids [25,26,27]. The antioxidant properties of tomato fruits mainly stem from the presence of polyphenols, including flavonoids and hydroxycinnamic acids, which possess the capability to effectively neutralize peroxyl radicals [28]. In humans, the development of several chronic diseases, including cardiovascular disease and atherosclerosis, is significantly influenced by oxidative stress, which arises from the disparity between the generation of reactive oxygen species (ROS) and the cellular ability to neutralize these harmful molecules [29]. It is important to note that antioxidant substances that maintain non-toxic ROS levels in the wound tissues could improve healing [30]. Furthermore, polyphenols extracted from the peel, pulp, and seeds of specific fruits, such as tomatoes, demonstrate an inhibitory effect on the proliferation of various cancer cell lines [30]. The lipophilic compounds found in tomatoes could provide a beneficial impact on human health, with carotenoids exhibiting apoptotic effects in cancer cells [31] and lycopene being shown to lower levels of oxidized LDL cholesterol [32].

NVs isolated from tomato fruits were recently found to be anti-inflammatory in lipopolysaccharide-stimulated monocytic THP-1 cells [33]. However, the wound healing effects of tomato-derived NVs (TDNVs) in in vitro models of representative cell types involved in wound healing, i.e., fibroblasts and keratinocytes, have not been investigated so far. Thus, the full potential of TDNVs as effectors of regenerative tissue capacities has not been exploited yet. Bokka and colleagues have intriguingly isolated TDNVs by the use of chemicals with an enrichment step by either ultracentrifugation or size exclusion chromatography [34]. In this study, we approached the isolation of TDNVs by avoiding chemicals and exploring their effect on wound healing, making a further advancement toward a more “green” approach to the safe application of PDNVs.

## 2. Results

### 2.1. Physico-Chemical Characterization of TDNVs

Once the pellet of the ultracentrifugation step was obtained (Figure 1A), it underwent physico-chemical characterization along with the Extract. Transmission electron microscopy (TEM) analysis showed the morphology, integrity, and size of nanovesicles constituting both samples (Figure 1B). The vesicles examined in the Extract and NV samples are nanosized particles with a similar mean diameter of 115 nm and 130 nm, respectively (Figure 1C). PDI_TEM_ values, i.e., the PDI value for the number-based size distribution obtained by TEM images and calculated according to Tkachenko et al. [35], were 1.07 for the Extract and 1.04 for NVs, indicating, as expected, a narrower size distribution for the NV sample, having been obtained after further purification steps. Zeta potential analysis showed that both vesicle samples exhibited a negative surface charge (−19.6 mV for NVs and −7.8 mV for the Extract) (Figure 1D). The protein content, as analyzed by the BCA method, was found to be 9408 ± 1127 μg/mL (*n* = 9 biological replicates) in the Extract and 567.7 ± 31.3 μg/mL (*n* = 5 biological replicates) in NVs. Since the protein content of the NV sample was lower than the Extract, it does not allow a correct preparation of the different concentrations of vesicles to be tested in various future experiments; henceforth, we have used the stock concentration obtained from the pellet weight/volume. Otherwise, for the Extract sample, we have always considered the calculated protein dosage.

### 2.2. The Effect of Tomato NVs and Extract on the Wound Healing Process

We investigated the effect of different concentrations (30–200 µg/mL) [18] of NVs (*w*/*v*) and Extract (protein concentration) on wound repair exerted by HUKE monolayers upon mechanical scratching. Both the Extract and the NVs accelerated wound closure in a dose-dependent manner relative to the control (Figure 2A,B). While untreated (control) cells closed the scratched area after 48 h, although not completely, the Extract, at the highest concentration (200 µg/mL), already accelerated the wound closure by 6 h, with the width of the wound further reducing 24 h after treatment with concentrations of 75, 100, and 200 µg/mL. Furthermore, in the presence of 200 µg/mL of the Extract, the wound completely closed after 48 h compared to the control, in which the wound remained open for approximately 12% (Figure 2A,C). Instead, NVs started to have a statistically significant effect on wound repair only after 24 h and at concentrations of 100 and 200 µg/mL. Also, in this case, the wound was almost completely closed after 48 h (Figure 2B,D).

We also investigated the effect of NVs and Extract on wound closure by injured NIH-3T3 fibroblast monolayers, another cell type involved in wound healing. Different from HUKE epithelial cells, fibroblasts were faster in closing the wound; indeed, the wound closure already had occurred at 24 h (Figure 3A,B). On the other hand, the Extract and NVs accelerated the wound closure already at 6 h, even though only with the highest concentrations, i.e., 100 and 200 µg/mL (Figure 3A,B), with a statistically significant difference as compared with respective controls (Figure 3C,D).

Overall, these data illustrate a pro-healing effect of both tomato NVs and Extract on two different cell types participating in wound closure, i.e., keratinocytes and fibroblasts, after injury. Since it is well known that cell proliferation and migration are both necessary components for wound closure, we investigated these two events.

### 2.3. Effects of Tomato NVs and Extract on Cell Proliferation

To investigate cell proliferation, we performed an MTT colorimetric assay at 24 h and 48 h. Figure 4A,B show that all concentrations up to 200 µg/mL for NVs and Extract exerted no cytotoxic effect on HUKE; however, additionally, no increasing effects were observed, indicating that neither NVs nor the Extract had any effect on keratinocyte proliferation. When the same assay was performed on NIH-3T3 cells, a similar behavior was observed (Figure 4C,D), again indicating that both NVs and Extract were not altering the proliferation of fibroblasts.

To observe whether injured cells treated with NVs or Extract were secreting some mediators exerting a proliferative effect, conditioned media were collected from NV- or Extract-treated cells and used in the MTT assay on the same cell types. Cells treated with an unconditioned medium were used as controls. No effect was observed when the conditioned medium from NV- or Extract-treated injured HUKE monolayers was tested on the proliferation of HUKE (Figure 4E,F). On the other hand, an inhibitory effect on NIH-3T3 cell proliferation was observed at 48 h when these cells were exposed to a conditioned medium obtained from NIH-3T3 injured monolayers and treated with different concentrations of NVs or Extract (Figure 4G,H). While the Extract was determining this effect at all concentrations (Figure 4G), NVs were effective at decreasing cell proliferation only at 200 μg/mL (Figure 4H).

In summary, NVs and Extract did not exert a direct effect on keratinocyte and fibroblast proliferation, while an indirect inhibitory effect was observed with fibroblasts, likely due to secreted mediators. Therefore, the acceleration of wound closure, previously achieved with the NVs and the Extract, could be due to something else, such as cell migration.

### 2.4. Effect of NVs and Extract on Cell Migration

To evaluate cell migration, we used the agarose spot assay [36,37]. NIH-3T3 fibroblasts were allowed to adhere to spot-containing dishes and were evaluated for their capacity to invade the agarose spot either containing EGF (as positive control) [38] or NVs/Extract. As testified by an increase in the distance of cell migration from the edge, chemotactic invasion of keratinocytes was observed following the addition of EGF to molten agarose (Figure 5A, last column), while migration was lower in the absence of EGF (Figure 5A, first column). Migration was time-dependent since the distance from the forefront to the edge was higher at 48 h than at 24 h. Figure 5A also shows that both NVs and Extract enhanced cell migration as compared to controls, although with lower efficiency than EGF. To understand whether NVs or Extract have an indirect effect, the conditioned medium from scratched monolayers, either untreated or treated with Extract/NVs, was collected after 24 h and embedded into the molten agarose. Results also show that the conditioned medium of cells treated with either Extract or NVs induced cell migration (Figure 5A). It is worth noting that the increase was statistically significant for all these conditions in comparison with untreated cells at both 24 and 48 h (Figure 5B,C).

HUKE migrated slower than NIH-3T3 cells, as evidenced by control and EGF-containing agarose spot (Figure 6A, first and last columns; compared with the same columns of Figure 5A), but with a net difference between these two conditions. NVs exerted their pro-migratory effect at both time points only with 200 μg/mL. On the other hand, the Extract gave an enhancing effect at 24 h with 200 μg/μL and with both concentrations at 48 h (Figure 6A), confirmed by the statistical analysis (Figure 6B). The agarose spot assay performed with conditioned media mirrored at 24 h the results obtained by adding directly NVs or Extract, with the only difference that the Extract had a significant effect at 48 h only with 200 μg/mL (Figure 6A,C).

## 3. Discussion

In this work, we have isolated and characterized nanometer-sized vesicles from cultivation spread in Apulia, *Solanum lycopersicum* (tomato), and evaluated TDNVs as facilitators of wound closure upon injury of keratinocytes and fibroblasts. PDNVs have attracted a lot of attention in medicine since they are derived from green natural products, and their use is highly sustainable. Indeed, PDNVs are suitable for bulk preparation due to their relatively simple source, often easily accessible plants [39]. To approach the possibility of using PDNVs in human applications, the presence of harsh chemicals should be avoided. In order to isolate TDNVs, an extraction buffer (pH 8) composed of phosphate, ethylenediamine tetraacetic acid (EDTA), and protease inhibitors was previously used [33,34]. Instead, we directly obtained juice from tomato fruits and subjected them to a simple differential centrifugation protocol. Based on our findings, the isolation of nanovesicles from tomato juice was successfully performed. In fact, the presence of nanoparticle structures was demonstrated in our samples of NVs and Extract obtained from this procedure. The scientific literature is rich in publications regarding the isolation of microvesicles (MVs), NVs, and apoplastic vesicles (AVs) of plant origin from different organs, such as fruit, flower, seed, rhizome, and leaf [40,41]. Investigations conducted in these samples by electron microscopy generally highlight the presence of heterogeneous vesicle-like objects with different shapes, including spherical, ellipsoidal, or cup-shaped morphologies and a wide size distribution, which are typical characteristics of this type of complex sample [34,42]. Although the average sizes of plant-derived NVs depend on the source matrix and on the purification method, these are in the range between 30 and 500 nm [11,39]. The presence of cup-like structures confirmed the morphology obtained by others in tomato-derived NVs [34]. Our values are, therefore, consistent with the size values typical of NVs extracted from plant matrices and, in particular, are similar to values previously reported by other authors for NVs extracted from tomatoes, i.e., 110 ± 10 nm [33,34]. Regarding zeta potential values, it is well known that all eukaryotic cells (including those from animals, plants, and fungi) have a negative potential. Plant-derived NVs also maintain a negative potential. Their surface charges, as measured by zeta potential, are, for example, around −12/−17 mV for ginger-derived NVs and −2/−39 mV for broccoli-derived NVs [11]. The values measured for our preparations are, therefore, perfectly in line with expectations and previous experimental observations. Thus, by using a simple differential centrifugation protocol, we isolated TDNVs, avoiding harsh chemicals and solvents.

TDNVs have been shown to reduce macrophage inflammatory response [33], reverse microbiota dysbiosis caused by *Fusobacterium nucleatum* [43], induce chondrogenic differentiation of adipose-derived stem cells [44], display antioxidant activity [45], and have been exploited as green platforms for drug delivery vehicles [33,45,46]. Although previous research was carried out on the pro-healing effects of PDNVs both in vivo [16,17,22] and in vitro [18,19,20,21,22], to the best of our knowledge, nobody has investigated the role of TDNVs in an in vitro wound closure setting. Keratinocytes and fibroblasts should migrate and proliferate to fulfill their specific functions in this context, i.e., re-epithelization and production of an extracellular matrix to convey appropriate migration of skin progenitor cells, respectively [47,48]. The scratch assay showed that NVs and the Extract accelerated the wound closure of injured monolayers of keratinocytes and fibroblasts, indicating an overall pro-healing effect. While both NVs and Extract gave an enhancing outcome on the migration of both cell types, no effects were observed on cell proliferation. It is worth noting that Extract was used as a 16.6-fold higher protein concentration in comparison to NVs, arguing that NVs display a stronger capacity than Extract to induce wound closure and cell migration as well, henceforth attributing this biological activity to NVs themselves. Whether these effects are given by distinct NV components (i.e., membrane phospholipids and proteins, intravesicular proteins, microRNAs, and metabolites) [49] is to be ascertained. Antioxidant substances present in ripe tomatoes could be effective in contributing to wound healing by restricting hyperoxidative stress and limiting the duration of the inflammation procedure [50,51]. Various antioxidative components, such as lycopene, β-carotene, ascorbic acid, total phenolics, vitamin E, and vitamin C, enriched in tomatoes, can effectively relieve the oxidative stress in the wound bed and accelerate the rate of wound enclosure when a tomato powder, likely containing NVs, is included into a hydrogel formulation [52]. Thus, both the Extract and NVs of this study may contain those compounds that, by employing their strong antioxidant properties, accelerate the rate of wound closure.

The conditioned medium of monolayers injured and treated with NVs/Extract exerted an enhancing outcome on cell migration, while an indirect inhibitory effect was observed only with proliferating fibroblasts. The following can be inferred: (1) keratinocytes and fibroblasts secrete mediators of migration of the same cell types; (2) fibroblasts are sensitive to some soluble mediator(s) produced during induction by Extract (less by NVs) at 48 h and capable of reducing their proliferation, arguing either for some anti-mitotic molecule(s) present in the Extract or that the Extract induces some anti-proliferative fibroblast mediators. For example, interferon-γ, an anti-fibrotic cytokine, can be secreted by fibroblasts [53] and has inhibitory effects on NIH-3T3 fibroblast proliferation [54] and migration [55]. This is reasonable if we think that fibroblasts already closed the wound at 24 h (Figure 3). Why this inhibitory effect does not occur for keratinocytes is still unknown, but it can be presumably attributed to inhibitory factors secreted by other cell types in the later stages of wound repair. Paracrine secretion of TGF-β by macrophages and fibroblasts (absent in our keratinocyte monocultures) [56] may induce a reversion of keratinocytes to the basal phenotype, thus arresting keratinocyte proliferation [57].

Keratinocyte migration occurs in the form of a stratified epithelial sheet with a burst of mitotic activity occurring in a distinct subpopulation of epidermal keratinocytes located behind the migrating epithelium; however, it is still disputed if basal or suprabasal cells are involved in the re-epithelialization process [48]. Also, epithelial–mesenchymal transition mechanisms have been invoked. Studies on more complex models (e.g., three-dimensional organotypic full-thickness in vitro skin wound models comprising keratinocytes and fibroblasts [58]) are warranted for an in-depth understanding of the NV role in this phenomenon [48]. Our future studies also aim to understand whether this effect on cell migration will be obtained on vascular endothelial cells and, hence, on neoangiogenesis.

How do we use PDNVs, and particularly TDNVs, in the context of wound healing amelioration? Important and relevant actual components of the wound healing process are now considered stem cells, growth factors, and decellularized dermal matrices, which can more closely recapitulate the natural regenerative healing process and overcome the present drawbacks of current therapies [59]. Tissue engineering skin substitutes provide both dermal and epidermal components and can protect open wounds, promote ingrowth of fibrovascular tissue, and suppress granulation tissue and scars, with each having its own limitations, such as reduced vascularization, poor mechanical integrity, failure to integrate, scarring, and immune rejection [60]. Better formulations should then be envisioned, among which are those bearing stem cells and/or their secretome, including EVs [61]. The application of stem cells’ secretome may avoid aberration linked to whole cells, for instance, tumor appearance. Furthermore, EVs and secretomes from stem cells are endowed with multiple capacities in wound healing, comprising angiogenesis, inflammation inhibition, and promotion of proliferation and migration of keratinocytes as well as of fibroblasts [62]. A topical application of an epidermal/dermal scaffold bearing TDNVs on the wound bed would be then tested in relevant in vivo models.

## 4. Materials and Methods

### 4.1. Isolation of Solanum lycopersicum (Tomato)-Derived Nanovesicles (TDNVs)

TDNVs were isolated from tomato juice. The method was performed in accordance with relevant institutional, national, and international guidelines and legislation guidelines and complied with the Convention on Biological Diversity (https://www.cbd.int/convention/, accessed on 17 February 2024) and the Convention on the Trade in Endangered Species of Wild Fauna and Flora (https://cites.org/eng, accessed on 17 February 2024). Piccadilly tomato fruits (~600 g), purchased from the local market in Foggia (Italy), were washed twice with distilled water and subsequently with 1× phosphate-buffered saline (PBS) (Corning, New York, NY, USA; Cat# 20-031-CV) to remove soil and pesticides. Then, tomatoes were ground at the highest speed for a few minutes in a juicer mixer (Centrika Slow Juicer metal 177/1, Ariete, Campi Bisenzio, Firenze, Italy; Cat# 00C017710AR) to obtain juice without seeds and skins. The collected sample was carefully homogenized in a potter elvehjem tissue homogenizer (30 cm^3^) and then subjected to low-speed differential centrifugations at 1000× *g* and twice at 10,000× *g* for 10 min for each step at 4 °C to remove intracellular organelles, fibers, cellular debris, and larger fragments of tissue. Consequently, some aliquots of the filtered extract (“Extract”) obtained were stored at −80 °C until use, while the remaining part of the supernatant was filtered through a 0.22 µm vacuum filter system and then ultracentrifuged (Optima L-90 K, Beckman Coulter, Fullerton, CA, USA) at 146,000× *g* for 90 min at 4 °C using a Type SW 32 Ti Beckman rotor. The recovered pellet was ultracentrifuged at 146,000× *g* for 60 min at 4 °C in PBS 1× and then collected and resuspended in PBS 1×. The weight of this pellet (“NVs”) was measured by subtracting the obtained weight of the tube containing the pellet from the weight of the empty tube. Finally, the suspensions were stored at −80 °C until use.

### 4.2. Physicochemical Characterization of Tomato NVs and Extract

The presence of nanoparticulate structures in the samples, their morphology, integrity, and size were analyzed by transmission electron microscopy (TEM), while vesicles Zeta Potential was measured by laser Doppler electrophoresis (LDE) using a Zeta sizer Nano ZS (Malvern Panalytical Ltd., Malvern, UK) and in diffusion barrier mode with water [63]. The TEM analysis was done by a method previously described [63]. Size statistical analysis (vesicles average size and size distribution) of each sample was performed on 250 nanostructures by means of a freeware Image J analysis program (National Institutes of Health, Bethesda, MD, USA). PDI_TEM_ values were calculated on TEM values according to Tkachenko et al. [35].

### 4.3. Determination of Protein Concentration

The protein quantification of the filtered tomato Extract and NV samples was performed using a Pierce™ Bicinchoninic acid (BCA) Protein Assay Kit (Thermo Scientific, Waltham, MA, USA; Cat# 23225).

### 4.4. Cell Cultures

The human keratinocyte cell line (HUKE) was obtained from the Istituto Zooprofilattico Sperimentale della Lombardia e dell’Emilia Romagna (Brescia, Italy) and was grown in EpiLife medium (GIBCO, Fisher Scientific Italia, Segrate, Milan, Italy) supplied with the Human Keratinocyte Growth Supplement (HKGS) at 37 °C with 5% CO_2_ and 1% penicillin. NIH mouse fibroblast cell line (NIH-3T3, a kind gift of Dr. Giuseppe Procino, University of Bari “A. Moro”, Bari, Italy) was grown in Dulbecco’s modified Eagle medium (DMEM) with 4.5 g/L glucose and sodium pyruvate (Corning), containing 10% fetal bovine serum (FBS), 1% L-glutamine, and 1% penicillin/streptomycin [64].

### 4.5. Scratch Assay

HUKE or NIH-3T3 cells were seeded onto 24-well plates at concentrations of 5 × 10^4^ and 4 × 10^4^ cells/well, respectively. Mechanical damage was induced on the monolayer of the two single cell lines by using a P10 pipette tip. Cells were exposed to escalating doses of NVs (weight/volume) and Extract (protein concentration) and then incubated in a humidified incubator for 24 h. Wound closure was assessed using the Leica DM IRB inverted microscope consisting of a Leica DFC450 C camera (Leica, Wetzlar, Germany). The wounds were photographed in the same place at various points and several times (t.0, and after 6 h, 24 h, and 48 h). The size of the wound after repair compared with the initial wound area at each time point was measured through the ImageJ software (v. 1.53e, National Institutes of Health, Bethesda, MD, USA). In some experiments, monolayers were injured and treated with NVs or Extract at all concentrations for 24 h; then, the conditioned medium was collected and used immediately for cell proliferation experiments. The same procedure was used for the cell migration assay but using only the conditioned medium collected following treatment of the injured cell monolayers with the highest doses of NVs or Extract (100 and 200 µg/mL) for 24 h. Control medium was obtained from monolayers scratched and untreated.

### 4.6. Cell Proliferation Assay

HUKE or NIH-3T3 cells were seeded onto 96-well plates at concentrations of 1 × 10^4^ and 3 × 10^3^ cells/well, respectively, and treated with NVs and Extract at various concentrations for 24 h and 48 h in a humidified incubator (5% CO_2_ in air at 37 °C). The cell proliferation was assessed with Methyl-thiazol-tetrazolium (MTT, Sigma-Aldrich, Milan, Italy; Cat# MKBL6157V), as previously performed [63]. Three independent experiments were performed to generate the percentage of growth versus control (untreated cells).

### 4.7. Agarose Spot Assay

The cell migration measurement by agarose spot assay was performed following the method of Wiggins and Rappoport [38]. Initially, the agarose was dissolved in PBS 1× at a concentration of 0.5% until boiling on a hot plate. When the solution reached the temperature of 40 °C, different agar spots were prepared containing EGF (0.03 mg/mL) as positive control, or they were prepared with NVs and Extract at the two highest concentrations (100 and 200 µg/mL) or the conditioned medium collected during the scratch assay after inducing damage to the cells and treating them with NVs and Extract for 24 h; the untreated culture medium was employed as negative control. Each sample was diluted 1:10 with agarose before forming the spot. Ten-microliter of each agarose spot were pipetted, using cut pipet tips, as rapidly as possible, onto 24-well plate and allowed to cool for ~5 min at 4 °C. HUKE or NIH-3T3 cells were seeded into spot-containing plate in the presence of 10% FBS cell culture media and incubated for 4 h to allow them to adhere. Cells were then transferred into the respective culture media containing 0.1% FBS, repositioned into the incubator, and finally analyzed by microscopy after 24 h. Images were acquired using a Leica DM IRB inverted microscope consisting of a Leica DFC450 C camera (Leica, Wetzlar, Germany). The straight distance from the border of the spot to the migration front of cells penetrated the agarose spot [65] was analyzed using ImageJ.

### 4.8. Statistical Analysis

Statistical analysis was performed using Prism for Windows, version 5.01 (Graph-Pad SoftwareInc., San Diego, CA, USA), based on one-way analysis of variance (ANOVA) with Tukey’s multiple comparison post hoc test. The reference value to be considered for statistically significant differences was *p* < 0.05. Data are shown as means ± SD.

## 5. Conclusions

In conclusion, according to our findings, the isolation of nanovesicles from tomato juice was successfully performed in the absence of chemicals and solvents, which is extremely advantageous for future applications. These NVs were shown to potentially play an important role in the wound healing process as they improve damage repair. Cell migration is the main mechanism induced by TDNVs. The role of extracellular vesicles, derived from mammalian and plant cells, in all four phases of wound healing (hemostasis, inflammation, proliferation, and remodeling), and thus their regenerative capacities, has been ascertained [10]. However, due to their limitations in vivo half-life [66], it is now clear that they should administered in combination with dermal substitutes, allowing their survival in the harsh wound environment and pumping out their molecule cargoes in eliciting a spatiotemporal correct wound healing process.

## Figures and Tables

**Figure 1 ijms-25-02452-f001:**
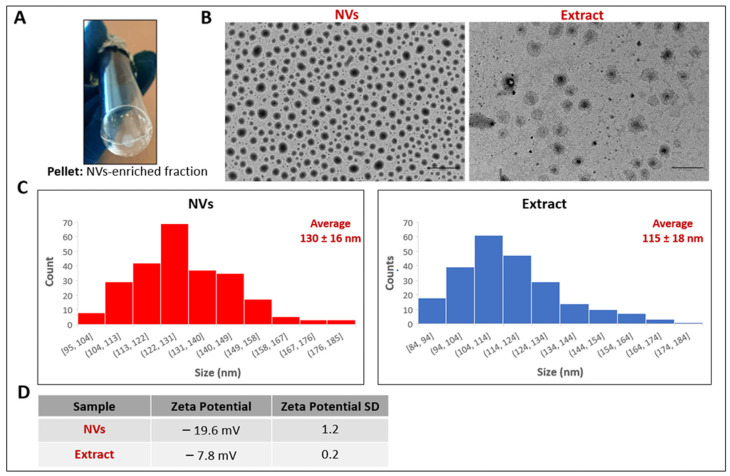
Identification and physicochemical characterization of tomato NVs and Extract. (**A**) NV white pellet obtained after ultracentrifugation at 146,000× *g* for 90 min at 4 °C. (**B**) Nanovesicles contained in NV and Extract samples were analyzed by TEM. The scale bars indicate 500 nm. (**C**) Size distribution measured on 250 vesicles for each sample using ImageJ program. (**D**) The surface zeta potential of particles was measured using a Zeta Sizer Nano ZS.

**Figure 2 ijms-25-02452-f002:**
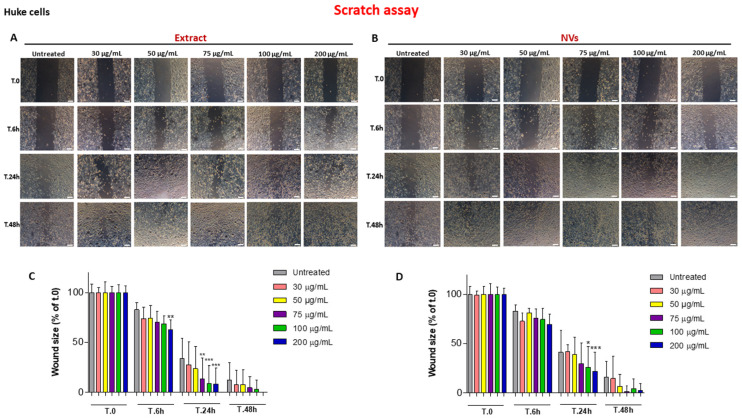
Wound closure with HUKE cells. (**A**,**B**) Images of HUKE monolayers at 0, 6 h, 24 h, and 48 h after the injury and the treatment with different concentrations (30–200 µg/mL) of the Extract (**A**) or NVs (**B**). Images of untreated are the same for A and B. Bar = 100 µm. (**C**,**D**) Percentages of wound size at different time points calculated relative to the percentage of the wound area at t.0 are considered 100%. The results are represented as mean ± SD of three experiments. * *p* < 0.05; ** *p* < 0.01; *** *p* < 0.0001 as compared to respective controls.

**Figure 3 ijms-25-02452-f003:**
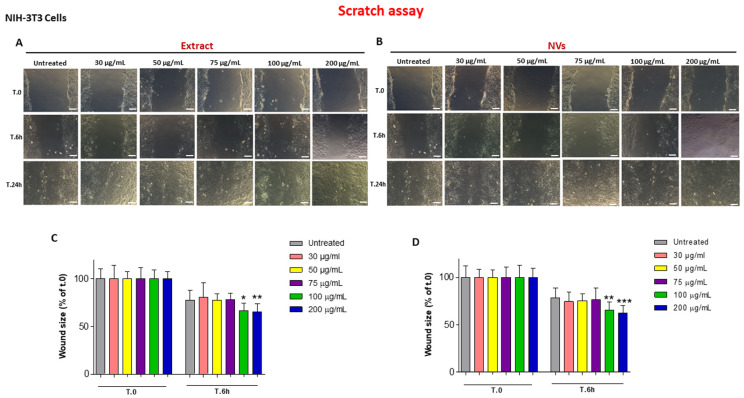
Wound closure with NIH-3T3 cells. (**A**,**B**) Images of NIH-3T3 monolayers at 0, 6 h, 24 h, and 48 h after the injury and the treatment with different concentrations (30–200 µg/mL) of the Extract (**A**) or NVs (**B**). Images of untreated are the same for A and B. Bar = 100 µm. (**C**,**D**) Percentages of wound size at different time points calculated relative to the percentage of the wound area at t.0 are considered 100%. The results are represented as mean ± SD of three experiments. * *p* < 0.05; ** *p* < 0.01; *** *p* < 0.0001 as compared to respective controls.

**Figure 4 ijms-25-02452-f004:**
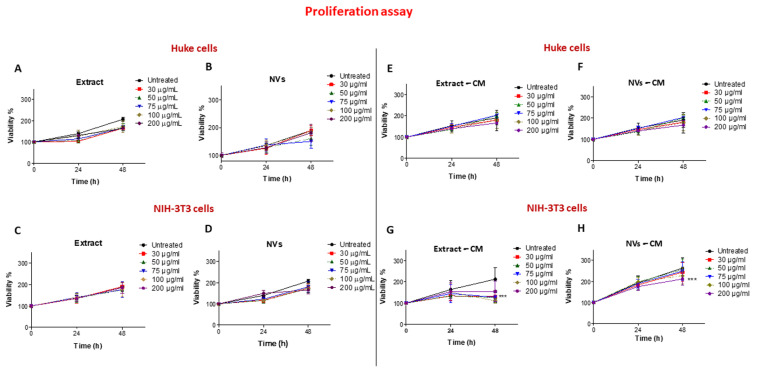
Effect of NVs and Extract on cell proliferation. (**A**–**D**) Proliferation of HUKE (**A**,**B**) and NIH-3T3 (**C**,**D**) was assessed by using the MTT colorimetric assay. The effect of different concentrations (30–200 µg/mL) of Extract (**A**,**C**) and NVs (**B**,**D**) on cell proliferation was assessed at 0, 24 h, and 48 h. Percentage of viability at different time points was calculated relative to the percentage of viability at t.0 and considered 100%. (**E**–**H**) Proliferation of HUKE (**E**,**F**) and NIH-3T3 (**G**,**H**) was assessed by using the MTT colorimetric assay. The effect of conditioned medium (CM) from Extract-treated injured monolayers (**E**,**G**) or from NV-treated injured monolayers (**F**,**H**) on cell proliferation was assessed at 0, 24 h, and 48 h. Percentage of viability at different time points was calculated relative to the percentage of viability at t.0 and considered 100%. Controls (untreated) are cells treated with unconditioned medium. The results are represented as mean ± SD of three experiments. *** *p* < 0.0001 vs. untreated cells.

**Figure 5 ijms-25-02452-f005:**
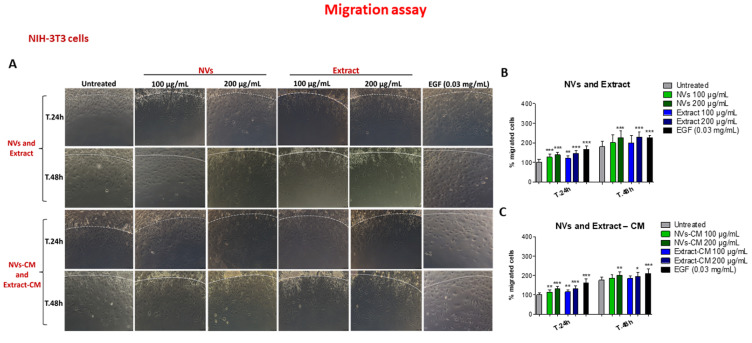
Effect of NVs and Extract on NIH-3T3 fibroblast migration. (**A**) Migration of NIH-3T3 cells in the agarose spot assay. Spots were left untreated, embedded with EGF, NVs, or Extract at 100 and 200 μg/mL, or conditioned medium (CM) obtained from injured monolayers treated with NVs/Extract, and observed by the microscope 24 h or 48 h later. The dotted white line indicates the spot border. Bar = 100 µm. (**B**,**C**) Percentage of migrated cells (distance of the forefront from the border) at different time points was calculated relative to the percentage of migration at t.0 and considered 100%. The results are represented as mean ± SD of three experiments. * *p* < 0.05; ** *p* < 0.01; *** *p* < 0.0001 as compared to untreated for each time point.

**Figure 6 ijms-25-02452-f006:**
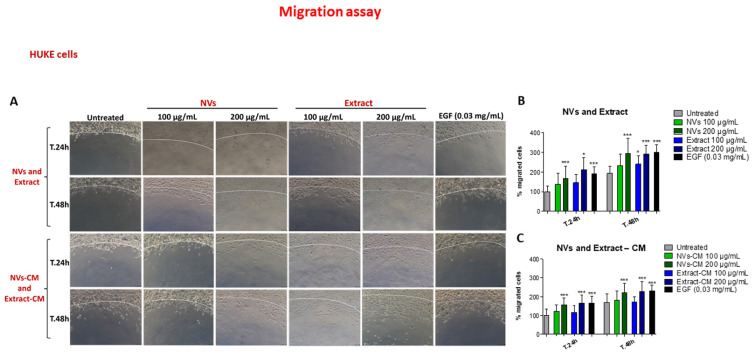
Effect of NVs and Extract on HUKE cell migration. (**A**) Migration of HUKE in the agarose spot assay. Spots were left untreated, embedded with EGF, NVs, or Extract at 100 and 200 μg/mL, or conditioned medium (CM) obtained from injured monolayers treated with NVs/Extract, and observed by the microscope 24 h or 48 h later. The dotted white line indicates the spot border. Bar = 100 µm. (**B**,**C**) Percentage of migrated cells (distance of the forefront from the border) at different time points was calculated relative to the percentage of migration at t.0 and considered 100%. The results are represented as mean±SD of three experiments. * *p* < 0.05; *** *p* < 0.0001 as compared to untreated for each time point.

## Data Availability

The authors will share data to support this study upon reasonable request.

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
