# Peer review of "Solanum lycopersicum (Tomato)-Derived Nanovesicles Accelerate Wound Healing by Eliciting the Migration of Keratinocytes and Fibroblasts"

_ijms, 2024, doi:10.3390/ijms25052452_

Round 1
Reviewer 1 Report
Comments and Suggestions for Authors
I have read the submission entitled 'Solanum lycopersicum (Tomato)-derived nanovesicles accelerate wound healing by eliciting migration of keratinocytes and fibroblasts'. The study is interesting and design well. Please consider my suggestion and revised it before publication.
1. In the Introduction section, authors introduced the advantages of bioactivities of vegtable or furite. However, even we classify them as vegetable, they are different plant with different active compounds. I suggest author add information about tomato rather than others.
2. use professional figure editing software to improve resolution, especially in bar.
3.Ref 29, is the tomato's advantage or NV's? I think those compounds both contain in the two samples.
4. text mistakes such as in line 320 and 327.
5. what is the reason author select NV?
Author Response
I have read the submission entitled 'Solanum lycopersicum (Tomato)-derived nanovesicles accelerate wound healing by eliciting migration of keratinocytes and fibroblasts'. The study is interesting and design well. Please consider my suggestion and revised it before publication.
We thank the Reviewer for her/his appreciation of our work.
1. In the Introduction section, authors introduced the advantages of bioactivities of vegtable or furite. However, even we classify them as vegetable, they are different plant with different active compounds. I suggest author add information about tomato rather than others.
A1. As suggested by the Reviewer, we have included in the Introduction a paragraph with more information about the advantages of bioactives in tomato fuits (lines 101-118).
2. use professional figure editing software to improve resolution, especially in bar.
A2. We have improved resolution at 600 dpi of all the Figures.
3.Ref 29, is the tomato's advantage or NV's? I think those compounds both contain in the two samples.
A3. Thanks for this observation. We added a sentence after this reference about the possibility that both Extract and NVs contain the compounds studied in Ref. 29 (now Ref. 52).
4. text mistakes such as in line 320 and 327.
A4. They were corrected.
5. what is the reason author select NV?
A5. NVs from mammalian and plant cells are at the forefront of medical advancements, and particularly in the context of wound repair, due to their interesting features and easy translability into the clinical arena. We are now including a novel paragraph illustrating the advantages brought by NVs, and for the purpose of this paper, by PDNVs (lines 75-85).
Reviewer 2 Report
Comments and Suggestions for Authors
Please downplay the significance of these results. For example, in the abstract, change 'NVs from tomatoes can accelerate wound healing' to 'NVs from tomatoes may accelerate wound healing'.
Also change 'NVs were shown to play an important role in the wound healing process as they improve damage repair' to 'NVs were shown to potentially play an important role in the wound 410 healing process as they improve damage repair'.
Comments on the Quality of English LanguageThere are multiple grammatical and spelling errors.
Author Response
Please downplay the significance of these results. For example, in the abstract, change 'NVs from tomatoes can accelerate wound healing' to 'NVs from tomatoes may accelerate wound healing'.
Also change 'NVs were shown to play an important role in the wound healing process as they improve damage repair' to 'NVs were shown to potentially play an important role in the wound 410 healing process as they improve damage repair'.
Thanks, we have made these changes.
Comments on the Quality of English Language
There are multiple grammatical and spelling errors
Thank you for advicing us on this issue, we have thoroughly checked the grammar and corrected spelling errors.
Reviewer 3 Report
Comments and Suggestions for Authors
The current experimental article is an interesting study on the development of Solanum lycopersicum (Tomato)-derived nanovesicles for wound healing. It is overall well-done, with several relevant assays having been performed. Hence, I only advise for the following alterations before acceptance for publication:
- In the introduction section, more should be said about wound healing, namely pathophysiology, current therapies and their limitations, why there is a need for therapeutical innovation, etc.;
- The obtained particle size and zeta potential (lines 86 to 89) should be discussed: were the results expected? And they considered good or not? Why? Specify in this context;
- If the authors used Zetasizer for the measurement of zeta potential, why not for particle size? Although TEM gives particle size results, it does not provide PDI values, which are always extremely valuable for the assessment of nanoformulation homogeneity; I suggest that the authors perform these measurements and add these results;
- Figure 1 quality (resolution) should be improved; it is currently quite difficult to read some parts; same with figures 4, 5 an 6;
- For better understanding, different colors should be added to the bars on the graphics of figure 2; same with figure 3;
- Given the promise of the obtained results, the authors should discuss on possible future formulation composition and administration route for the developed formulations;
- The conclusion section should be extended;
- An abbreviation list is missing and should be added.
Author Response
The current experimental article is an interesting study on the development of Solanum lycopersicum (Tomato)-derived nanovesicles for wound healing. It is overall well-done, with several relevant assays having been performed. Hence, I only advise for the following alterations before acceptance for publication:
We thank the Reviewer for her/his appreciation of our work.
- In the introduction section, more should be said about wound healing, namely pathophysiology, current therapies and their limitations, why there is a need for therapeutical innovation, etc.;
A1. We have now introduced more about the pathophysiology of wound healing, current therapies and their limitations, considering thus the need for therapeutical innovation (lines 39-65).
- The obtained particle size and zeta potential (lines 86 to 89) should be discussed: were the results expected? And they considered good or not? Why? Specify in this context;
A2. The scientific literature is rich in publications regarding the isolation of microvesicles (MV), nanovesicles (NV) and apoplastic vesicles (AV) of plant origin from different organs, such as fruit, flower, seed, rhizome and leaf. Investigations conducted in these samples by Electron Microscopy generally highlight the presence of heterogeneous vesicle-like objects with different shapes and a wide size distribution which are typical characteristics of this type of complex sample [https://doi.org/10.3390/foods9121852]. Although the average sizes of plant-derived NVs depend on the source matrix and on the purification method, these are in the range between 30 and 500 nm [DOI: 10.1039/c7tb03207b].
Therefore, our values are consistent with the size values typical of NVs extracted from plant matrices and in particular are similar to values previously reported by other authors for NVs extracted from tomatoes [https://doi.org/10.3390/foods9121852 - https://doi.org/10.3390/pharmaceutics15020333].
Regarding zeta potential values, it is well known that all eukaryotic cells (including those from animals, plants, and fungi) have a negative potential. Plant-derived NVs also maintain a negative potential. Their surface charges, as measured by zeta-potential, are for example around -12/-17 mV for ginger and -2/ -39 mV for broccoli derived NVs [DOI: 10.1039/c7tb03207b]. The values measured for our preparations are therefore perfectly in line with expectations and previous experimental observations.
These clarifications regarding both sizes and zeta potentials were added to the revised version of the manuscript, as suggested by the Reviewer, at lines 313-331 of the Discussion.
- If the authors used Zetasizer for the measurement of zeta potential, why not for particle size? Although TEM gives particle size results, it does not provide PDI values, which are always extremely valuable for the assessment of nanoformulation homogeneity; I suggest that the authors perform these measurements and add these results;
A3. The DLS analysis of the samples were carried out using the Malvern Zetasizer instrument, but size distribution and PDI values were not reported in the submitted manuscript because the software returned unstable values and often a negative "Size quality report". DLS analysis belongs to the so-called “indirect methods”, it analyzes the nanoparticles that make up the sample collectively and, by applying statistical methods, provides a size distribution based on the intensity of the light scattering signal. Samples that do not rigorously respect some characteristics (sphericity of the nanoparticles, perfectly known refractive index of the particles and of the dispersing medium, absence of spectral interference in absorption and emission with the laser source of the instrument, tendency to sediment, etc.) are not suitable to be analyzed with this technique [1]. Probably due to the same experimental difficulties, other authors who isolated NVs from tomato showed very broad size distributions and did not report the PDI value [2-3].
To overcome this difficulty, we opted for a "direct method", such as Transmission Electron Microscopy (TEM), for the determination of both the nanoparticle morphology and the size distribution. TEM is a “direct method” that returns a “number-based” size distribution and is not only applicable to nanopharmaceutical preparations but is also considered by some authors to be the best method [4]. This is because the images obtained provide information on the presence of aggregates or agglomerates in the sample; in addition in the number-based size distribution, the contribution of the larger nanoparticles which are often not representative of the sample is not overestimated as in DLS analysis.
The PDI can be calculated also in “direct methods” from the statistical analysis of the observed particles, and several formulas have been proposed in the literature [5-7] In the revised version of the manuscript, we have added the PDITEM, i.e. the PDI value for the number-based size distribution obtained by TEM images and calculated according to Tkachenko et al. [7].
[1] https://macro.lsu.edu/HowTo/MALVERN/PDF/MALVERN_FAQ_OTHER/Size%20quality%20report%20for%20the%20Zetasizer%20Nano.pdf
[2] Bokka et al., Biomanufacturing of Tomato-Derived Nanovesicles Ramesh, Foods 2020, 9, 1852; doi:10.3390/foods9121852
[3] Mammadova, R. et al., Protein Biocargo and Anti-Inflammatory Effect of Tomato Fruit-Derived Nanovesicles Separated by Density Gradient Ultracentrifugation and Loaded with Curcumin. Pharmaceutics 2023, 15, 333. https://doi.org/10.3390/pharmaceutics15020333
[4] Takechi-Haraya et al et al. Current Status and Challenges of Analytical Methods for Evaluation of Size and Surface Modification of Nanoparticle-Based Drug Formulations. AAPS PharmSciTech 23, 150 (2022). https://doi.org/10.1208/s12249-022-02303-y
[5] Stollenwerk, M. et al., Albumin-based nanoparticles as magnetic resonance contrast agents: I. Concept, first syntheses and characterisation. Histochem Cell Biol 133, 375–404 (2010). https://doi.org/10.1007/s00418-010-0676-z
[6] Midekessa G. et al., Zeta Potential of Extracellular Vesicles: Toward Understanding the Attributes that Determine Colloidal Stability, ACS Omega 2020, 5, 16701−16710 https://dx.doi.org/10.1021/acsomega.0c01582
[7] Tkachenko V. et al., Characterizing the Core-Shell Architecture of Block Copolymer Nanoparticles with Electron Microscopy: A Multi-Technique Approach, Polymers 2020, 12, 1656; doi:10.3390/polym12081656
- Figure 1 quality (resolution) should be improved; it is currently quite difficult to read some parts; same with figures 4, 5 an 6;
A4. The resolution of Figs. 1,4, and 6 has been brought to 600 dpi.
- For better understanding, different colors should be added to the bars on the graphics of figure 2; same with figure 3;
A5. Different colors were added to bars on the graphics of Figs. 2 and 3.
- Given the promise of the obtained results, the authors should discuss on possible future formulation composition and administration route for the developed formulations;
A6. These considerations have been added to the final part of the Discussion (lines 383-398).
- The conclusion section should be extended;
A7. The conclusion section has been extended.
- An abbreviation list is missing and should be added.
A8. An abbreviation list has been added at the end of the manuscript before the References.
Round 2
Reviewer 1 Report
Comments and Suggestions for Authors
All comments were addressed in the revised version.